# *ARID1A* Mutations Are Associated with Increased Immune Activity in Gastrointestinal Cancer

**DOI:** 10.3390/cells8070678

**Published:** 2019-07-04

**Authors:** Lin Li, Mengyuan Li, Zehang Jiang, Xiaosheng Wang

**Affiliations:** 1Biomedical Informatics Research Lab, School of Basic Medicine and Clinical Pharmacy, China Pharmaceutical University, Nanjing 211198, China; 2Cancer Genomics Research Center, School of Basic Medicine and Clinical Pharmacy, China Pharmaceutical University, Nanjing 211198, China; 3Big Data Research Institute, China Pharmaceutical University, Nanjing 211198, China

**Keywords:** *ARID1A* mutations, gastrointestinal cancer, cancer genomics, tumor immunity, tumor immunotherapy response

## Abstract

Because traditional treatment strategies for advanced gastrointestinal (GI) cancers often have a limited therapeutic effect, immunotherapy could be a viable approach for the therapy of advanced GI cancers, considering the recent success of immunotherapy in treating various refractory malignancies, including the DNA mismatch repair-deficient GI cancers. However, only a subset of cancer patients currently respond to immunotherapy. Thus, it is important to identify useful biomarkers for predicting cancer immunotherapy response. The tumor suppressor gene *ARID1A* has a high mutation rate in GI cancers and its deficiency is correlated with the microsatellite instability (MSI) genomic feature of cancer. We investigated the correlation between *ARID1A* mutations and tumor immunity using three GI cancer genomics datasets by the bioinformatic approach, and found that diverse antitumor immune signatures were more highly enriched in *ARID1A*-mutated GI cancers than in *ARID1A*-wildtype GI cancers. The elevated immune activity in *ARID1A*-mutated GI cancers was associated with the higher tumor mutation burden and lower tumor aneuploidy level, as well as a higher proportion of MSI cancers in this GI cancer subtype. Moreover, we found that *ARID1A*-mutated GI cancers more highly expressed *PD-L1* than *ARID1A*-wildtype GI cancers. The elevated antitumor immune signatures and *PD-L1* expression could contribute to the more active immunotherapeutic responsiveness and better survival prognosis in *ARID1A*-mutated GI cancers than in *ARID1A*-wildtype GI cancers in the immunotherapy setting, as evidenced in three cancer cohorts receiving immunotherapy. Thus, the *ARID1A* mutation could be a useful biomarker for identifying GI cancer patients responsive to immunotherapy.

## 1. Introduction

Gastrointestinal (GI) cancers are prevalent and account for a large number of cancer deaths globally [1]. Traditional treatment strategies for advanced GI cancers often have a limited therapeutic effect [2]. With the recent success of immunotherapy in treating various refractory malignancies [3,4,5,6], the immunotherapy strategy has become a viable approach for the therapy of advanced GI cancers [7]. A notable example is that two immune checkpoint inhibitors, pembrolizumab and nivolumab, have been clinically used for treating DNA mismatch repair-deficient GI cancers [7]. Despite these successes of cancer immunotherapy, a large proportion of cancer patients failed to respond to such therapy. Abundant evidence indicates that cancer immunotherapy response is associated with certain genetic or genomic features, such as PD-L1 expression [8], DNA mismatch repair deficiency [9], tumor mutation burden (TMB) [10], and tumor aneuploidy [11]. In addition, the mutation of specific genes in cancer may have a correlation with cancer immunotherapy response, e.g., the positive correlation of *TP53* and *KRAS* mutations with immunotherapy response in lung cancer [12]. 

ARID1A (AT-rich interaction domain 1A) is a component of the ATP-dependent chromatin remodeling complex SNF/SWI which is involved in transcriptional activation of genes normally repressed by chromatin. *ARID1A* is frequently mutated in a wide variety of cancers [13] and its mutation is associated with a poor prognosis in certain cancers such as liver cancer [14] and breast cancer [15]. Previous studies have shown that *ARID1A* mutations were often mutually exclusive with *TP53* mutations in cancer, indicating that ARID1A may also play an important role in tumor suppression [16]. Many studies have explored the association of *ARID1A* deficiency with GI cancers [17,18,19,20] and revealed that the *ARID1A* loss was associated with poor prognosis in GI cancers [17,21].

A number of studies have demonstrated that *ARID1A* deficiency may promote antitumor immunity, as well as *PD-L1* expression [16,22,23]. The positive correlation of *ARID1A* deficiency with tumor immunity has been attributed to the elevated tumor mutation load caused by the deficiency of DNA mismatch repair that is regulated by ARID1A [16]. In this study, using three publicly available GI cancer genomics datasets [24,25,26], we explored the correlation of *ARID1A* mutations with tumor immunity in GI cancers by a bioinformatic approach. We confirmed that *ARID1A* mutations are associated with elevated immune activity in GI cancers and demonstrated that the increased tumor mutation load and reduced tumor aneuploidy in *ARID1A*-mutated GI cancers may explain the more active immune signatures in *ARID1A*-mutated GI cancers versus *ARID1A*-wildtype GI cancers. Furthermore, we found that *ARID1A*-mutated cancers likely exhibited a better survival prognosis than *ARID1A*-wildtype cancers in the immunotherapy setting. The reason behind this observation could be that the elevated immune activity and *PD-L1* expression promote the immunotherapy response in *ARID1A*-mutated cancers. This study is a bioinformatic analysis based on publicly available datasets, which furnishes new insights into the association between *ARID1A* deficiency and antitumor immunity in GI cancers.

## 2. Results

### 2.1. ARID1A Mutations Are Associated with Elevated Immune Activity in GI Cancers

We found that diverse immune signatures (CD8+ T cells, NK cells, immune cytolytic activity, activated CD4+ T cells, activated dendritic cells, MHC class I) consistently showed significantly higher enrichment levels in *ARID1A*-mutated GI cancers than in *ARID1A*-wildtype GI cancers (Mann-Whitney U test, *P* < 0.05) (Figure 1A). Moreover, we compared the ratios between the mean expression levels of immune-stimulatory signature marker genes versus immune-inhibitory signature marker genes. We found that the ratios (CD8+ T cells versus CD4+ regulatory T cells, pro-inflammatory cytokines versus anti-inflammatory cytokines, and immune-promoting M1 macrophages versus immune-inhibiting M2 macrophages) were significantly higher in *ARID1A*-mutated GI cancers than in *ARID1A*-wildtype GI cancers (Mann-Whitney U test, *P* < 0.05) (Figure 1B). These results again suggest that *ARID1A* mutations are associated with elevated immune activity in GI cancers. In addition, we found that a number of human leukocyte antigen (HLA) genes were upregulated in *ARID1A*-mutated GI cancers relative to *ARID1A*-wildtype GI cancers (Student’s *t* test, *P* < 0.05) (Figure 1C), suggesting the higher immunity in *ARID1A*-mutated GI cancers.

Furthermore, we found that numerous immune-related KEGG [28] pathways were significantly upregulated in *ARID1A*-mutated GI cancers versus *ARID1A*-wildtype GI cancers by GSEA [27]. These pathways included allograft rejection, antigen processing and presentation, cytokine-cytokine receptor interaction, cytosolic DNA-sensing, graft-versus-host disease, leishmania infection, natural killer cell mediated cytotoxicity, NOD-like receptor signaling, RIG-I-like receptor signaling, T cell receptor signaling, and Toll-like receptor signaling (Figure 1D). These results again demonstrate that *ARID1A* mutations are associated with increased immune signatures in GI cancers.

### 2.2. ARID1A Mutations Are Associated with Increased TMB and Reduced Tumor Aneuploidy Levels in GI Cancers

TMB positively correlates with tumor immune signatures and immunotherapy response [10,29], while tumor aneuploidy levels negatively correlate with them [11]. We observed that *ARID1A*-mutated GI cancers showed significantly higher TMB (defined as the total number of somatic mutations in tumor) than *ARID1A*-wildtype GI cancers (Mann-Whitney U test, *P* < 0.001) (Figure 2A). In contrast, *ARID1A*-mutated GI cancers exhibited significantly lower aneuploidy levels than *ARID1A*-wildtype GI cancers (Mann-Whitney U test, *P* < 0.05) (Figure 2A). The significant correlations of *ARID1A* mutations with TMB and tumor aneuploidy levels may explain why *ARID1A*-mutated GI cancers have elevated immune activity compared to *ARID1A*-wildtype GI cancers.

To correlate *ARID1A* mutation, TMB, and aneuploidy level with immune activity in GI cancers, we built a logistic regression model with the three predictors (*ARID1A* mutation, TMB, and aneuploidy level) for predicting immune cytolytic activity in GI cancers. As expected, both *ARID1A* mutation and TMB were positive predictors and the aneuploidy level was a negative predictor for immune cytolytic activity (Figure 2B). In all three GI cancer cohorts, *ARID1A* mutation showed a positive prediction potential for immune cytolytic activity (β coefficient: β = 1.08, 0.64, 1.07, and *P* = 0.047, 0.061, 0.068 for STAD-1, STAD-2, and COAD, respectively). These data again suggest that *ARID1A* mutations are associated with increased immune signatures in GI cancers.

### 2.3. ARID1A Mutations Are Associated with the Microsatellite Instability (MSI) Genomic Feature of GI Cancers

A previous study has shown that ARID1A can promote mismatch repair (MMR) by interacting with MMR protein MSH2 [16]. As a result, ARID1A deficiency may lead to impaired MMR and thus correlates with the MSI genomic feature of cancer [16]. Indeed, we observed a significant positive correlation between the *ARID1A* mutation and the MSI genomic feature in all three GI cancer cohorts (Fisher’s exact test, *P* < 0.001) (Figure 3A). Moreover, in both TCGA GI cancer cohorts, the *ARID1A* mutation was significantly co-occurring with the mutation of MMR genes, including *MSH2*, *MSH6*, *MLH1*, *PMS2*, *MSH3*, and *MLH3* (Fisher’s exact test, *P* < 0.001) (Figure 3B). These results confirmed that *ARID1A* deficiency is associated with the MSI genomic feature and impaired MMR in cancer [16]. As expected, MSI-high (MSI-H) GI cancers had significantly higher TMB than MSI-low (MSI-L) and microsatellite stable (MSS) GI cancers (Mann-Whitney U test, *P* < 0.001) (Figure 3C). In contrast, MSI-H GI cancers had significantly lower aneuploidy levels than MSI-L/MSS GI cancers (Mann-Whitney U test, *P* < 0.001) (Figure 3C). Furthermore, we observed that MSI-H GI cancers exhibited significantly stronger immune signatures than MSI-L/MSS GI cancers (Mann-Whitney U test, *P* < 0.01) (Figure 3D). This is consistent with previous studies showing that MSI is associated with high immunogenic activity in cancer [9]. Collectively, these data suggest that *ARID1A* mutations lead to MMR deficiency or MSI thereby contributing to the elevated tumor immunity in GI cancers.

In addition, we compared the enrichment levels of immune signatures between *ARID1A*-mutated and *ARID1A*-wildtype GI cancers within the MSI-L/MSS subtype. We found that the correlations between *ARID1A* mutations and tumor immune signatures were significantly weaker in the MSI-L/MSS subtype than in all GI cancers in STAD-2 and COAD (Appendix A). However, all these immune signatures still showed significantly higher enrichment levels in *ARID1A*-mutated STAD-1 than in *ARID1A*-wildtype STAD-1 within the MSS subtype (Mann-Whitney U test, *P* < 0.001) (Appendix A). The immune cytolytic activity, MHC class I, and activated CD4+ T cells were more highly enriched in *ARID1A*-mutated MSI-L/MSS STAD-2 than in *ARID1A*-wildtype MSI-L/MSS STAD-2 (Mann-Whitney U test, *P* < 0.05). The ratios of pro-/anti-inflammatory cytokines and M1/M2 macrophages were significantly higher in *ARID1A*-mutated MSI-L/MSS STAD-2 than in *ARID1A*-wildtype MSI-L/MSS STAD-2 (Mann-Whitney U test, *P* < 0.05) (Appendix A). These results suggest that some other factors beyond MSI could contribute to the elevated tumor immunity in *ARID1A*-mutated GI cancers as well.

### 2.4. ARID1A Mutations Are Associated with Elevated PD-L1 Expression in GI Cancers and Favorable Immune Checkpoint Blockade Therapy Response in Cancer

We found that *ARID1A*-mutated GI cancers had significantly higher *PD-L1* expression levels than *ARID1A*-wildtype GI cancers (Student’s *t* test, *P* < 0.01) (Figure 4A). This is consistent with previous studies showing that ARID1A deficiency is associated with elevated *PD-L1* expression in GI cancers [23]. Furthermore, we explored the correlation between *ARID1A* mutations and immune checkpoint blockade therapy response using three cancer immunotherapy (anti-PD-1/PD-L1/CTLA-4) response-associated cohorts (Allen cohort [3], Hugo cohort [31], and Samstein cohort [32]). We compared the immunotherapy response rates between *ARID1A*-mutated and *ARID1A*-wildtype cancers (melanoma) in the Allen cohort and Hugo cohort and found that *ARID1A*-mutated cancers had higher response rates than *ARID1A*-wildtype cancers (42.86% versus 25.81% in Allen cohort, and 100% versus 51.43% in Hugo cohort) (Figure 4B). Moreover, we compared the overall survival (OS) between *ARID1A*-mutated and *ARID1A*-wildtype cancers in the three cohorts. In the Samstein cohort (pan-cancer), *ARID1A*-mutated cancers had a significantly better OS than *ARID1A*-wildtype cancers (log-rank test, *P* = 0.005) (Figure 4C). Moreover, *ARID1A*-mutated GI cancers had a significantly better OS than *ARID1A*-wildtype GI cancers in the Samstein cohort (log-rank test, *P* = 0.031) (Figure 4C). In the Allen and Hugo cohorts, *ARID1A*-mutated cancers had favorable OS trends compared to *ARID1A*-wildtype cancers (log-rank test, *P* = 0.095 and 0.333 for Allen and Hugo cohorts, respectively) (Figure 4C). Furthermore, we examined the correlation between *ARID1A* mutations and OS in individual cancer types in the Samstein cohort and found that *ARID1A* mutations were associated with a better OS in head and neck cancer (log-rank test, *P* = 0.01) (Appendix A). Appendix A also shows that *ARID1A* mutations were associated with more favorable OS trends in melanoma, lung cancer, breast cancer, esophagogastric cancer, and colorectal cancer (log-rank test, *P* < 0.25). Nevertheless, the correlation between *ARID1A* mutations and OS was unlikely to be significant in the three GI cancer cohorts (STAD-1, STAD-2, and COAD) not receiving immunotherapy (Figure 4D). These data indicate that *ARID1A*-mutated cancer patients respond favorably to immune checkpoint blockade therapy versus *ARID1A*-wildtype cancer patients and therefore are likely to have a better OS prognosis. The more active immunotherapeutic responsiveness in *ARID1A*-mutated cancer patients could be attributed to the elevated tumor immunity and PD-L1 expression in this cancer subtype.

## 3. Discussion

Our bioinformatic analysis revealed that *ARID1A* mutations had a significant positive association with tumor immunity in GI cancers. This significant association was attributed to the dysfunction of ARID1A in regulating MMR proteins that results in deficient MMR and MSI genomic features in GI cancers. Besides the elevated antitumor immune signatures, *ARID1A*-mutated GI cancers exhibited significantly higher *PD-L1* expression levels than *ARID1A*-wildtype GI cancers that would enhance the immunotherapy response and accordingly result in a better survival prognosis in this cancer subtype. Thus, the *ARID1A* mutation could be a predictive biomarker for the response to anti-PD-1/PD-L1 immunotherapy, as evidenced by the present and other studies [16,23,33]. 

Besides a number of immune-related pathways, many cancer-associated pathways were upregulated in *ARID1A*-mutated GI cancers versus *ARID1A*-wildtype GI cancers, identified by GSEA [27]. These cancer-associated pathways included apoptosis, cell cycle, DNA replication, and p53 signaling. The associations between these pathways and ARID1A have been extensively investigated [34,35,36,37]. Interestingly, all these pathways have been significantly associated with tumor immunity [38,39,40,41]. Altogether, these data suggest the intertwined relationship between ARID1A deficiency, ARID1A-mediated pathways, and tumor immunity.

The MSI genomic feature is a key factor responsible for the elevated immunity in *ARID1A*-mutated GI cancers. However, we found that immune signature scores were still significantly higher in *ARID1A*-mutated than in *ARID1A*-wildtype GI cancers within the MSI-L/MSS subtype (Appendix A). It suggests that other factors beyond MSI, such as the reduced tumor aneuploidy level and deregulation of ARID1A-mediated pathways, may also contribute to the increased tumor immunity in *ARID1A*-mutated GI cancers (Figure 5). 

There are several limitations in this study. First, we only used publicly available datasets in this study. The use of proprietary datasets to confirm the findings obtained from the analyses of public datasets would improve this study. Second, although we obtained the novel finding that *ARID1A* mutations had a significant negative correlation with tumor aneuploidy levels, experimental validation could be necessary to confirm this finding from bioinformatic analysis. It would be a priority for our future studies. 

## 4. Conclusions

The *ARID1A* mutation is associated with the elevated immune activity and *PD-L1* expression in GI cancers and could be a useful biomarker for identifying GI cancer patients responsive to immunotherapy.

## 5. Methods

### 5.1. Materials

Three GI cancer genomics datasets were utilized in this study, including STAD-1 (the gastric cancer genomics dataset from the Asian Cancer Research Group (ACRG) [24]), STAD-2 (the gastric cancer genomics dataset from The Cancer Genome Atlas (TCGA) [25]), and COAD (the colon cancer genomics dataset from TCGA [26]). The STAD-1 gene expression profiling data was obtained from NCBI Gene Expression Omnibus (GEO) (GSE62254) and its gene somatic mutation and clinical data was downloaded from the publication [24]. The STAD-2 and COAD datasets were downloaded from the GDC data portal (https://portal.gdc.cancer.gov/). In addition, we obtained three cancer genomics datasets (Allen cohort [3], Hugo cohort [31], and Samstein cohort [32]) containing anti-PD-1/PD-L1/CTLA-4 immunotherapy response-associated clinical data from their associated publications. 

### 5.2. Comparisons of Immune Signature Enrichment Levels between Two Groups of Samples

The enrichment level of an immune signature (represented by a set of marker genes) in a sample was quantified by the single-sample gene-set enrichment analysis (ssGSEA) score [42]. A total of six immune signatures were analyzed, including CD8+ T cells, NK cells, immune cytolytic activity, activated CD4+ T cells, activated dendritic cells, and MHC class I. Their marker genes were obtained from two publications [43,44]. We compared the enrichment levels of immune signatures between two groups of samples using the Mann-Whitney U test. We also compared the ratios between immune-stimulatory signatures and immune-inhibitory signatures (CD8+/CD4+ regulatory T cells, pro-/anti-inflammatory cytokines, and M1/M2 macrophages) between two groups of samples on the basis of the mean expression levels of immune-stimulatory/inhibitory signature marker genes. The marker genes of all these immune signatures are listed in Appendix A. We performed these analyses in R programming environment (R package “GSVA” for calculating ssGSEA scores and R function “wilcox.test" for performing the Mann-Whitney U test).

### 5.3. Gene-Set Enrichment Analysis

Based on the three GI cancer gene expression profiling datasets (STAD-1, STAD-2, and COAD), we used GSEA [27] to identify three sets of KEGG [28] pathways that were significantly upregulated in *ARID1A*-mutated GI cancers relative to *ARID1A*-wildtype GI cancers using a threshold of FDR < 0.05. The pathways common in all three pathway sets were defined as the significantly upregulated pathways in *ARID1A*-mutated GI cancers versus *ARID1A*-wildtype GI cancers.

### 5.4. Evaluation of Tumor Aneuploidy Levels

For each tumor sample, we used the ABSOLUTE algorithm [30] to calculate its ploidy score that represented the tumor aneuploidy level.

### 5.5. Prediction of Tumor Immune Activity Using ARID1A Mutation, TMB, and Tumor Aneuploidy Level 

We used logistic regression to assess the contributions of *ARID1A* mutation, TMB, and tumor aneuploidy levels in predicting tumor immune activity. The three predictors *ARID1A* mutation (mutated or wildtype), TMB (defined as the total somatic mutation count in tumor), and tumor aneuploidy level (defined as the tumor ploidy score generated by ABSOLUTE [30]) were binary, discrete, and continuous variables, respectively. The tumor immune cytolytic activity (high (upper third) versus low (bottom third)) was predicted. We performed the logistic regression analysis in the R programming environment using the R function “glm” to fit the binary model. We calculated the standardized regression coefficients (β values) using the function “lm.beta” in R package “QuantPsyc”.

### 5.6. Survival Analyses

We compared OS between *ARID1A*-mutated and *ARID1A*-wildtype cancers in the three GI cancer datasets and the three cancer immunotherapy response-associated datasets. Kaplan-Meier survival curves were used to show the survival differences and the log-rank test was used to evaluate the significance of survival time differences. We performed the survival analyses using R programming function “survfit” in “survival” package.

## Figures and Tables

**Figure 1 cells-08-00678-f001:**
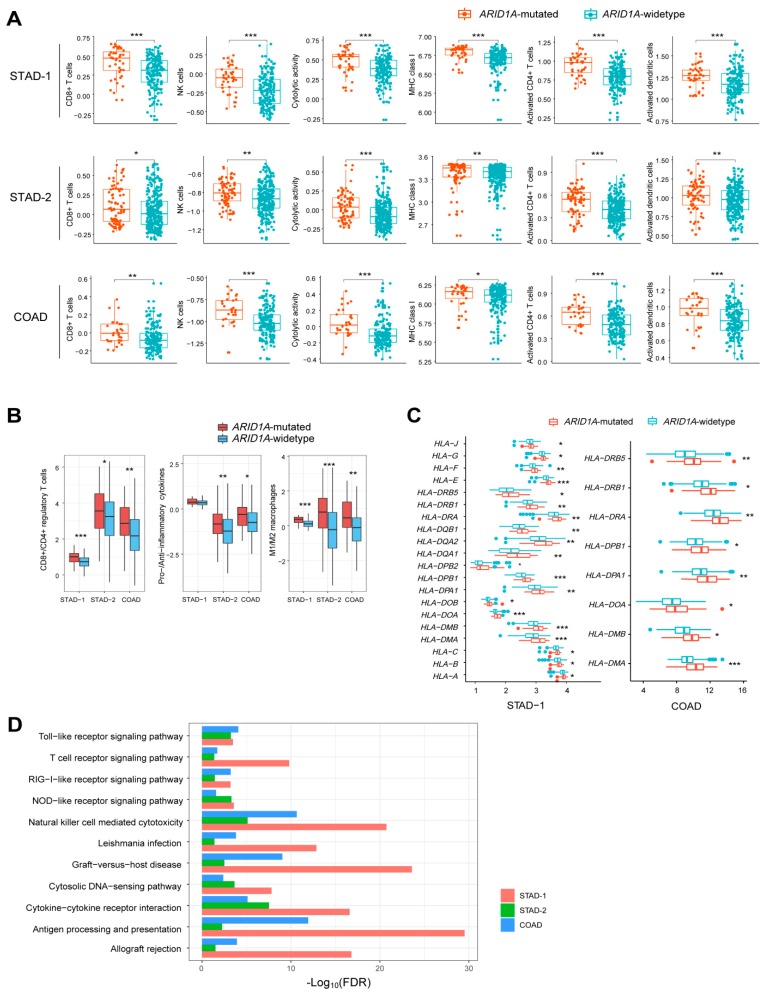
*ARID1A*-mutated gastrointestinal (GI) cancers have elevated immune activity compared to *ARID1A*-wildtype GI cancers. (**A**) Diverse immune signatures show significantly higher enrichment levels (ssGSEA scores) in *ARID1A*-mutated GI cancers than in *ARID1A*-wildtype GI cancers (Mann-Whitney U test, *P* < 0.05). (**B**) The ratios between immune-stimulatory signatures and immune-inhibitory signatures are significantly higher in *ARID1A*-mutated GI cancers than in *ARID1A*-wildtype GI cancers (Mann-Whitney U test *P* < 0.05). (**C**) A number of human leukocyte antigen (HLA) genes are more highly expressed in *ARID1A*-mutated GI cancers than in *ARID1A*-wildtype GI cancers (Student’s *t* test, *P* < 0.05). (**D**) GSEA [27] identifies numerous immune-related KEGG [28] pathways which are more highly enriched in *ARID1A*-mutated GI cancers than in *ARID1A*-wildtype GI cancers. FDR: false discovery rate. * *P <* 0.05, ** *P <* 0.01, *** *P <* 0.001. STAD-1: The gastric cancer genomics dataset from the Asian Cancer Research Group (ACRG) [24]. STAD-2: The gastric cancer genomics dataset from The Cancer Genome Atlas (TCGA) [25]. COAD: The colon cancer genomics dataset from TCGA [26]. They also apply to following figures.

**Figure 2 cells-08-00678-f002:**
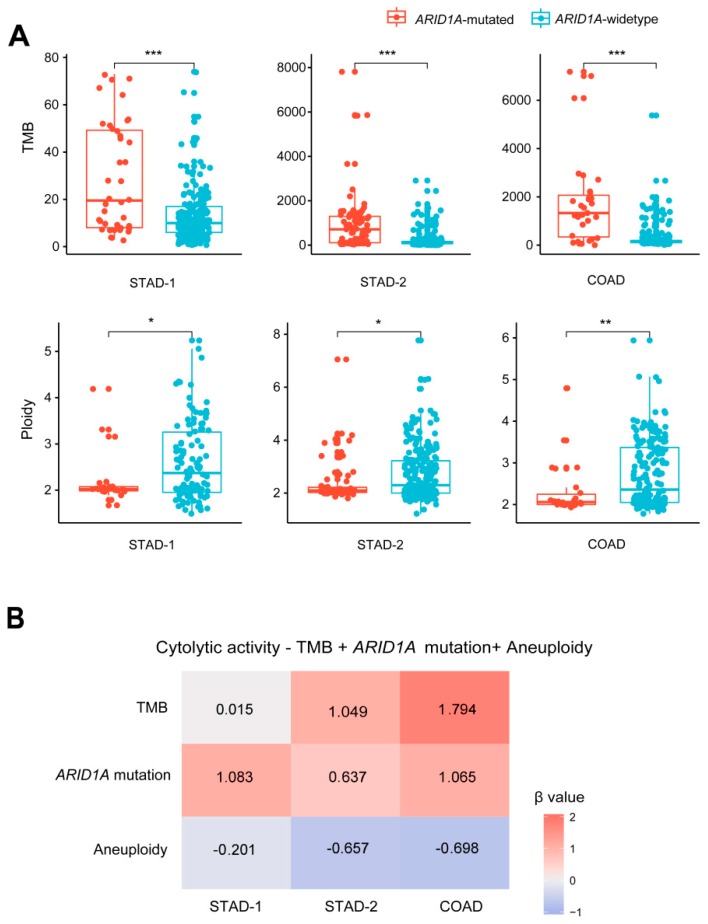
Associations among *ARID1A* mutations, tumor mutation burden (TMB), tumor aneuploidy levels, and immune activity in GI cancers. (**A**) *ARID1A*-mutated GI cancers have significantly higher TMB and lower aneuploidy levels than *ARID1A*-wildtype GI cancers (Mann-Whitney U test, *P* < 0.05). * *P <* 0.05, ** *P <* 0.01, *** *P <* 0.001. (**B**) Logistic regression analysis shows that *ARID1A* mutation and TMB are positive predictors and the aneuploidy level is a negative predictor for immune cytolytic activity. TMB is defined as the total number of somatic mutations in tumor and tumor aneuploidy level is defined as the tumor ploidy score assessed by ABSOLUTE [30].

**Figure 3 cells-08-00678-f003:**
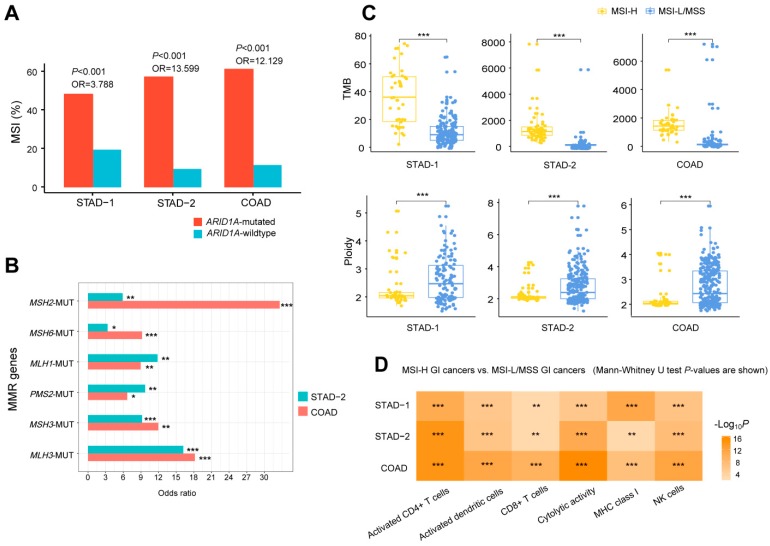
*ARID1A* mutations are associated with the microsatellite instability (MSI) genomic feature of GI cancers. (**A**) *ARID1A*-mutated GI cancers harbor a significantly higher proportion of MSI cancers than *ARID1A*-wildtype GI cancers (Fisher’s exact test, *P* < 0.001). OR: odds ratio. (**B**) The *ARID1A* mutation significantly co-occurs with the mutation of mismatch repair (MMR) genes (Fisher’s exact test, *P* < 0.001). MUT: mutation. (**C**) The MSI-high (MSI-H) GI cancers have significantly higher TMB and significantly lower aneuploidy levels than the MSI-low (MSI-L) and microsatellite stable (MSS) GI cancers (Mann-Whitney U test, *P* < 0.001). (**D**) MSI-H GI cancers have significantly stronger immune signatures than MSI-L/MSS GI cancers (Mann-Whitney U test, *P* < 0.01). * *P <* 0.05, ** *P <* 0.01, *** *P <* 0.001.

**Figure 4 cells-08-00678-f004:**
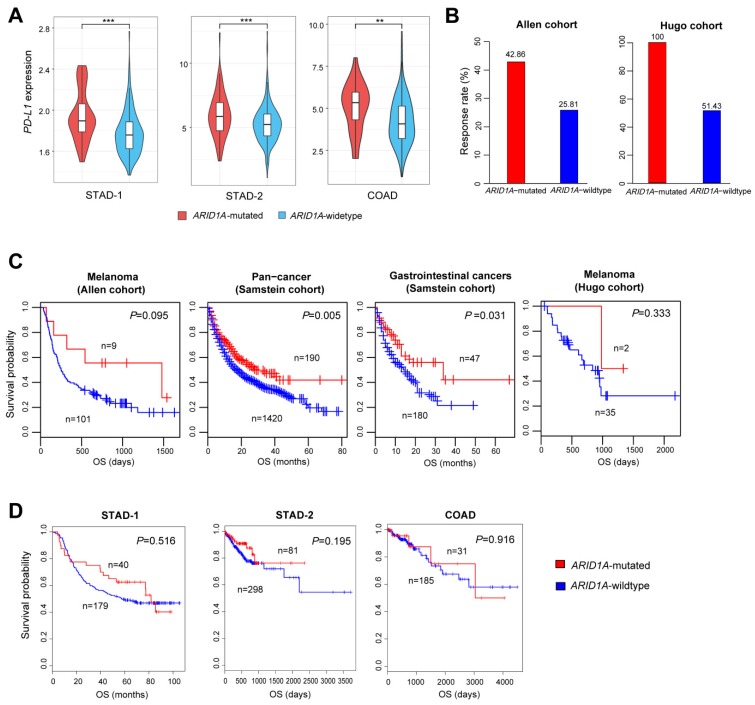
*ARID1A* mutations are associated with elevated *PD-L1* expression in GI cancers and favorable immunotherapy response in cancer. (**A**) *ARID1A*-mutated GI cancers have significantly higher *PD-L1* expression levels than *ARID1A*-wildtype GI cancers (Student’s *t* test, *P* < 0.01). * *P <* 0.05, ** *P <* 0.01, *** *P <* 0.001. (**B**) *ARID1A* mutations are associated with increased immune checkpoint blockade therapy response rate in two melanoma cohorts (Allen cohort [3] and Hugo cohort [31]). (**C**) Kaplan-Meier survival curves show that *ARID1A*-mutated cancers have a better overall survival (OS) than *ARID1A*-wildtype cancers in the immunotherapy setting in three cohorts (Allen cohort [3], Hugo cohort [31], and Samstein cohort [32]). (**D**) Kaplan-Meier survival curves show that the *ARID1A* mutation exhibits no significant correlation with OS in the three GI cancer cohorts (STAD-1, STAD-2, and COAD) untreated with immunotherapy.

**Figure 5 cells-08-00678-f005:**
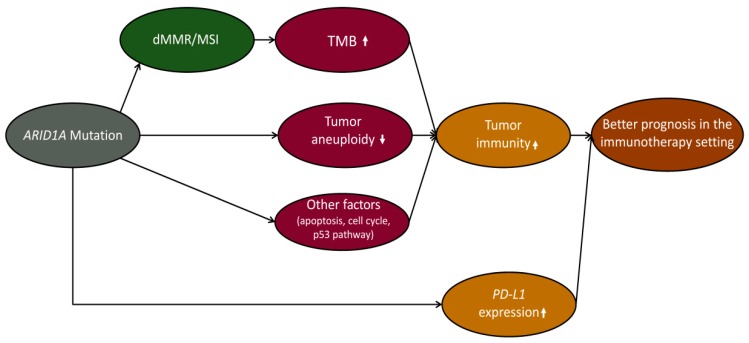
The mechanism by which the *ARID1A* mutation contributes to the elevated tumor immunity and tumor immunotherapy response, as well as favorable clinical outcomes in cancer patients receiving anti-PD-1/PD-L1/CTLA-4 immunotherapy.

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
