# Peer review of "ARID1A Mutations Are Associated with Increased Immune Activity in Gastrointestinal Cancer"

_cells, 2019, doi:10.3390/cells8070678_

Reviewer 1 Report

The manuscript titled " ARID1A Mutations Correlate with Increased Immune Activity in Gastrointestinal Cancer" is an extremely important topic and authors have attempted to present an interesting bioinformatic analysis of publicly available genomic data.

Major comments:

The authors have presented that ARID1A Mutations  signature could be used as biomarker in gastrointestinal cancer patients responsive to immunotherapy. However, authors should also analyze human GI cancer samples and demonstrate that ARID1A Mutational load have an impact in immunotherapy drug response. 

Authors have tried to repeat the same work which has been reported by Shen et al, 2018, Nature Medicine, volume 24, pages556–562. 

Thus based to the journal guidelines authors should complement the bioinformatic analysis with wetlab experimental data.

Authors should also demonstrate experimentally that ARID1A Mutation alleviate tumor aneuploidy in GI cancer.

Author Response

1. The authors have presented that ARID1A Mutations signature could be used as biomarker in gastrointestinal cancer patients responsive to immunotherapy. However, authors should also analyze human GI cancer samples and demonstrate that ARID1A Mutational load have an impact in immunotherapy drug response. 

Response: We thank the reviewer for this suggestion. We have tried to compare the immunotherapy response rates between ARID1A-mutated and ARID1A-wildtype cancers in human GI cancer samples, but could not find such data publicly available. Alternatively, we compared the overall survival (OS) between ARID1A-mutated and ARID1A-wildtype GI cancers in a cancer cohort (Samstein cohort [31]) receiving anti-PD-1/PD-L1/CTLA-4 immunotherapy and found that ARID1A-mutated GI cancers had a significantly better OS than ARID1A-wildtype GI cancers in this cohort (log-rank test, P=0.031) (see Figure 4C). However, the correlation between ARID1A mutations and OS was not significant in the cancer cohorts not receiving immunotherapy (see Figure 4D). It suggests that ARID1A mutations have an impact in immunotherapy drug response in GI cancers. In addition, we have compared the immunotherapy response rates between ARID1A-mutated and ARID1A-wildtype cancers in two human melanoma cohorts (Allen cohort [3] and Hugo cohort [30]) and found that ARID1A-mutated cancers had a higher response rate than ARID1A-wildtype cancers (42.86% versus 25.81% in Allen cohort, and 100% versus 51.43% in Hugo cohort) (see Figure 4B).       

2. Authors have tried to repeat the same work which has been reported by Shen et al, 2018, Nature Medicine, volume 24, pages556–562. Thus based to the journal guidelines authors should complement the bioinformatic analysis with wetlab experimental data.

Response: We thank the reviewer for this suggestion. Before we read the publication (Shen et al, 2018, Nature Medicine, volume 24, pages556–562), we have independently found the associations of ARID1A mutations with tumor immunity and tumor immunotherapy response by bioinformatics analysis and planned to perform wetlab experiments to validate our bioinformatics findings. However, when we found that a large amount of experimental data on this topic have been published by Shen et al, we quitted the plan of performing wetlab experiments. Instead, we focused on the bioinformatics analysis to add novel data on this topic. Certainly, we agree that some additional experimental data may add value in this study.

3. Authors should also demonstrate experimentally that ARID1A Mutation alleviate tumor aneuploidy in GI cancer.

Response: This is a valuable suggestion. We have showed that the negative correlation between ARID1A mutations and tumor aneuploidy may partially explain the positive correlation between ARID1A mutations and tumor immunity in GI cancers since the elevated tumor aneuploidy levels could result in the reduced tumor immunity. Therefore, the experimental validation of ARID1A mutations alleviating tumor aneuploidy would be of great significance. Unfortunately, we cannot complete such an experiment under current timeline and experimental condition. However, it would be top priority for our future studies.   

Reviewer 2 Report

I read with great interest the manuscript by Li et al. The manuscript is interesting and well conducted. Although the findings are quite important, the main limitation of the study is that the Authors used a series of data obtained by public dataset and not a proprietary set of GI patients. This should be explained in the text. Additionally, the software used for statistical analysis should be indicated. Minor orthographical and grammatical errors have been found throughout the manuscript.

Author Response

1. Although the findings are quite important, the main limitation of the study is that the Authors used a series of data obtained by public dataset and not a proprietary set of GI patients. This should be explained in the text.

Response: We thank the reviewer for this suggestion. We have added an explanation of this issue in the Discussion section.

2. Additionally, the software used for statistical analysis should be indicated.

Response: We thank the reviewer for this suggestion. We have added more description of the software used for statistical analysis in the Methods section.

3. Minor orthographical and grammatical errors have been found throughout the manuscript.

Response: We thank the reviewer for this comment. We have checked the manuscript thoroughly and adjusted orthographical and grammatical errors.
